# Corporealism as an Ontological Position and Its Involvement in the Thought of Tertullian

Marián Ambrozy

Department of Social Sciences, College of International Business ISM Slovakia in Prešov, 080 01 Prešov, Slovakia; ambrozy.marian@gmail.com

**Abstract:** This paper aims to examine the meaning, role, inspirations, and place of corporealism in Tertullian's system of thought. The extent to which corporealism is a basic philosophical belief in Tertullian's work and to what extent it is a particular element of his theological doctrine is questioned. It presents the named ontological position as a rare specificity within the range of early Christian thought, especially in Tertullian's works *De anima* and *De carne Cristi*. This paper makes a clear distinction between corporealism and materialism, as it tries to determine the degree of influence of Stoic philosophy, especially ontology, on Tertullian, as well as the influence of Aristotle in selected areas. In this context, his traducianism is also examined. In the ontological context, the status of the soul and God in Tertullian thought is also presented. In connection with the metaphysical problem of creation, the article also touches on the question of *creatio ex nihilo* as a problem on which Tertullian had to take a stand. It investigates the role of corporealism in Tertullian's polemic against Marcion, Apelles, and the Valentinian Gnosis by mapping which elements in the teachings of these representatives and Gnosis, especially (but not exclusively) Valentinian, could provoke Tertullian to controversy. This paper holds the opinion that Tertullian's corporealism was due to his theological views and controversy with opponents, which were used as philosophical inspiration, especially stoic inspiration, but was used mainly in the service of his theological thinking and strategic needs for argumentation.

**Keywords:** Tertullian; corporealism; stoa; Marcion; Apelles; traducianism; corporeality



## 1. Introduction

The old Stoic notion that a thing exists only when it acts itself or accepts action has resonated in the history of thought for a very long time. In practice, it means the corporeality of a being. Basically, the ontological moment (identification with changes with the body) appears in a weaker ontological version from the stoic of Zenon, through Hobbes' notion of the body, to the absolutization of the approach in identifying all existence with matter and its manifestation in Lenin, especially in his book *Materialism and Empiriocriticism*. The vulgar identification of materialism and atomism, and possibly other similar gross materializing distortions in the interpretation of pre-Socratics, is rejected as lacking historical meaning. They are intended for some (not only) Marxist-oriented historians of philosophy. The remarkable thing is that the mentioned ontological tendency also appears in early Christian thought. Representatives of patristics show a range of diverse approaches to the holistic teaching of the Christian image of the world. One of them is also the mentioned corporealism, which declares everything that exists to be bodily.

Tertullian is a representative of the mentioned ontological position in early Christian philosophy. This article will attempt to present his representation of the subject through the prism of the historical situation in which he found himself, primarily in terms of influencing previous currents of (theological and philosophical) thought, but especially in terms of controversies with ideological antipodes. The priority texts on which it will build are *The Treatise on the Soul* and *On the Flesh of Christ*. Let us recall that the Latin-speaking theologian

was not a common occurrence during Tertullian's lifetime, as the cultural language of the early Church was Greek. The first known Latin church writer was Marcus Minucius Felix. The question of whether Tertullian identified himself as an African under the Roman yoke was taken over by Wilhite (2007). Kitzler believes that the addressees of Tertullian's writings were the Christian community rather than anti-Roman dissidents (Kitzler 2008, p. 301). Given the appeal of researchers of recent decades, the montanistic card will not be taken into account with respect to Tertullian. According to several modern interpreters, e.g., Dunn (2007), it makes no sense to divide Tertullian's publications into pre-Montanist and Montanist periods.

The aim of this paper is to examine the context, purpose, and the nature of corporealism as a specific ontological position in the selected texts of Tertullian. Its role is to ascertain to what extent was it related to ontological reasons for postulating his own ontological position, and to what extent his reasons were different. In essence, the principal goal is to find out why Tertullian reached this ontological solution in particular.

This question is being studied in the context of the ideological opponents that Tertullian faced. These are chiefly Marcion, Apelles, and the Valentinian Gnosis. Polemic reactions to their opinion are also featured in the treatise *On the Flesh of Christ.* On the basis of Tertullian's polemic with these three ideological antipodes, the role of corporealism is established by attempting to identify specific motives that, in the work of Tertullian, evoked a polemic and a reaction in the form of his corporealistic ideological position. The degree of inspiration by the Stoics is likewise examined.

A further aspect of the research is the subject of the cosmogonic relations connected to Tertullian's corporealism. In this respect, this paper also explores the motive behind Tertullian's belief in creatio ex nihilo in opposition to the dominant notion in the Hellenized world of order arising from something, as well as the theological implications of Tertullian's utilization of this cosmogonic position. The focus is also placed on the benefits that this position brings about in his argumentation with his opponents.

The opinions of Tertullian's main ideological opponents are also taken into account through the prism of his views on the Body of Christ and cosmogony. These are well-known opponents featured in *On the Flesh of Christ*; however, the focus will be put on their thoughts regarding the Body of Christ and their cosmogony. These opinions are put into contrast with the ones of Tertullian, and theological consequences are also derived. Against the background of these consequences, the questions of the extent to which Tertullian's ontological position is a real ontological conviction and the extent to which it is about the purposeful use of philosophical instruments in favor of theological argumentation are answered. The results are presented in the discussion.

## 2. Corporealism and the Soul in Tertullian's Concept

*The Treatise on the Soul* is a space in which our author categorically rejects the dual perception of the soul and body. According to Daniel-Hughes, this is an attack on the Platonic dualism of both the material and immaterial (Daniel-Hughes 2011, p. 66). It is neither the first Christian systematics nor the first Christian psychology, as Kitzler points out (Kitzler 2015a, p. 44). He advocates for strict monism in the sense of corporealism. In *The Treatise on the Soul*, Tertullian expresses his polemic to those who are the opponents of its bodily nature. He means, in particular, Plato and Aristotle with their epigones. At the same time, the designation "mythical" indicates a clear disagreement with the authors who (at least in their understanding) consider the soul to be composed "of things obviously material" (Tertullianus 1950, p. 188). It attributes different elements to them, in accordance with different interpretations (especially of peripatetic provenance). Among them, we find traditional water, fire, and atoms, as well as the less-frequented blood and the fifth element. The Carthaginian theologian erred a little, as Kitzler points out, confusing the names Hipparchus and Hippasus (Kitzler 2018, p. 22). Tertullian speaks with understanding about Zeno of Citium and Cleanthes; the former says that after death we are left with a bodily spirit intertwined with a living being, while the latter speaks of similar mental signs

among relatives, according to which the soul is something bodily. This argument of the Stoics is welcomed by our philosopher. Although the effect on the soul and the body is seemingly separate, the suffering of both is perceived simultaneously (an injury to the body is also felt by the soul). This also applies vice versa. Tertullian also argues with the attitude of the Stoic logician Chryssipus of Soli, who argues that nothing disembodied can leave something corporeal that would apply to the soul. "Hence, we see that the soul is corporeal; unless it were, it could not be separated from the body" (Tertullianus 1950, p. 189). As Kitzler points out, "Tertullian's belief in the corporeality of the soul, or rather that the soul must have its body (corpus), is part of his corporeal ontology, the main thesis of which is that everything that is different from a real and independent being must be physical" (Kitzler 2015b, p. 46).

From a historical point of view, these are obvious elements of Stoicism in Tertullian's ontological approach. As Hladký points out, "the suffering principle is a precursor that has no category, no quality. What gives it its quality is an active principle identified with reason (Logos), that is, God (Deus)" (Hladký 2020, p. 24). The Stoic god shapes himself through matter; he is only an active aspect of one substance. The aspects of Stoicism in Tertullian's ontology are dealt with in an individual section.

Tertullian takes over a lot from the abovementioned ontological position. As in Stoicism, God has a physical essence. If this were not the case, it would not exist. He literally states this in *Adversus Praxean*, saying that God is a spirit, which is a body of a specific kind. Stoic inspiration implies much, not only for Tertullian's ontology, but also for his own solution to the psychophysical question. The soul, like other manifestations of the substance, is set in motion from the outside. He cites divine ecstasy as an example. Tertullian says that its role is also inverse, as it sets the body in motion. "It is the soul which moves the feet in walking, the hands in touching, the eyes in seeing, and the tongue in speaking" (Tertullianus 1950, p. 190). Finally, Tertullian argues that the soul is the mover of the body. As a disembodied mover, it would be ineffective. Kitzler points out here that the essence of anti-Platonist arguments against the dualism of the soul and the body is taken from Stoicism (Kitzler 2015a).

*2.1. Tertullian's Psychology*

The issue is directly related to the theory of perception. Tertullian did not bypass this question either. He argues with the position of the Platonists, who claim that the bodily and intellectual senses are separate, from which position they associate the incorporeality of the soul. According to the Platonists, the senses of the body speak of bodily entities, while the intellectual senses speak of intellectual entities, as reported by Tertullian. According to him, non-bodily entities are also subject to the bodily senses, namely the sound of hearing, the color of sight, and the fragrance of smell (*Ecce enim incorporalia ostendo corporalibus sensibus subici, sonum auditui, colorem conspectui, odorem odoratui...*) (Tertullianus 1950, p. 190). The senses of the body thus approach something disembodied in Tertullian's argument. He also refers to the doctor Soran, according to whom the soul is nourished by bodily things. If the body is not nourished by food, the soul leaves the body (Goldstein 1969). Tertullian adds that if the soul is about to leave the body, it will not be saved by receiving the thoughts of Plato and Aristotle, even though it feeds on disembodied entities. On this occasion, the attentive reader will not miss the fact that the Carthaginian writer points to the fact that philosophy can become entangled in its own specific language to the point of incomprehensibility "through a misunderstanding of their own doctrines" (Tertullianus 1950, p. 192). The possibility of the presence of several bodies in one place cannot be denied—in any philosophical dispute—as the birth of several children within one birth serves as an example. Tertullian cites the birth of living paternity as an authentic contemporary example.

Tertullian argues with the inherent authority of Scripture when he refers to the story of the rich man and Lazarus. As Kitzler points out, Irenaeus does not regard this story as allegorical, but as a report (Kitzler 2018, p. 34). Here, he argues about whether it is necessary

to interpret the story allegorically or to treat it as a literal interpretation. He refers to the fact that in both cases the soul in the underworld is punished by flames, since even in the allegorical case it is allegorized by the body, so our theologian concludes that it has a body. In the underworld, it must have some identity and it cannot be null, so it must be corporeal. If the soul is to receive something in terms of patience, it needs to be physical.

In the matter of the soul's suffering in terms of acceptance, the acceptance of punishment or reward obviously also has a place. According to Vysoký, this was also significant to the apologist Justin, specifically for his lost work *Περί ψυχής*, which he considers to be an important source for Tertullian (Vysoký 1937, p. 88).

Apparent contradictions do not necessarily mean exclusion. Tertullian argues that even in Empedocles there are opposing forces of love and strife, and from these opposites all individualities arise. If the soul is invisible, it does not mean the exclusion of its corporeality. In addition, its invisibility is relative, as the spirit saw it, and St. John saw the souls of the martyrs (*Sic Iohannes in spiritu dei factus animas martyrum conspicit*) (Tertullianus 1950, p. 196).

The soul is thus the body of a specific provenance; however, as Kitzler says, here our author is a bit entangled in his own argument. He is forced to attribute other bodily qualities to the soul; however, elsewhere he says that the soul is "sui generis, thus does not have to share the qualities of other bodies" (Kitzler 2018, p. 36). They are represented by the external form, boundedness, and three-dimensionality (Tertullianus 1950, p. 196). He refers to a kind of Christian woman with the gift of prophecy who perceived the soul in sight. He argues that the bodily substance of the soul has a form that is similar to the body in which a particular soul resides. It is Paul's term "inner man". As Kitzler points out, Tertullianus does not understand Paul's reversal ethically, but anthropologically (Kitzler 2018, p. 38). The author points out that he could have taken over the argument from Irenaeus.

## 2.2. Tertullian's Ontology

It should be noted that there is no correct identification of corporealism and materialism. These are two different ontological approaches. As Kitzler points out, corporeality is a conditio sine qua non for being; its opposite is not to be intangible, but non-existent, since corporeality is equal to the highest ontological category (Kitzler 2015a, p. 48). It is from this point of view that it is possible to perceive the medieval critique of Tertullian as a materialist as inappropriately addressed (Svennson 2020).

Kitzler tries to consider what Tertullian's ulterior motive might have been to take on corporealist positions. His method of justification can be described as reasonable. He argues that corporealism appeared only in polemical works of the Carthaginian theologian directed against various heretics (especially those who may be described as Gnostics) who were influenced by Platonism (in the context of the period, mainly by Platonism) (Kitzler 2018, p. 42). The abovementioned author has repeatedly pointed out the fact that Tertullian was an ardent person of dispute and often uses mutually incompatible arguments due to changes in the composition of the audience. As Torres points out (Torres 2013), Tertullian's polemical strategy was rebuttal and conviction. Sometimes his argument is bizarre; as Kaufman points out, he rejects the right of heretics to invoke the Bible (Kaufman 1991). According to Kitzler, he had been advocating for corporealism since the writing of *Adversus Hermogenem*. Thus, his work also exhibits a certain development of ideas. In the *Apologeticum*, he claims that the body will necessarily be present at the resurrection, because the soul cannot suffer without the body. The corporeal soul reunites with the body at the resurrection. "The eschatological revelation announced by Christ would not have taken place, rationally speaking, if the resurrected body could not be reunited in an intimate way with the immortal soul, because the final judgment of the individual soul is over actions of the same, done by will and mental decision but by the tools of the body" (Moreira Alaníz 2013, p. 64). If one is not pardoned, the soul will suffer along with the body (Tertullianus 1977). Thus, we perceive that Tertullianus, in terms of the entire body of his work, shows a heterodox view during its development.

Similarly, the influence of Stoicism on his work, especially on his perception of physics and metaphysics, can be perceived in a heterodox manner. Kitzler believes that it is not right to call him a Heraclitic stoic (Kitzler 2009, p. 168). Osborn came up with this idea, pointing out that Heraclitus also had considerable authority over the older early Christian authors Justin and Clement of Alexandria (Osborn 1998, p. 362). Despite the clear adoption of Stoic metaphysics in many works, we can also observe an attempt on his part to distance himself from Stoicism as long as it is contextually appropriate for his argument. In *De praescriptione hereticorum*, he says: let those who are Stoic, Platonic, or Dialectic Christians overtake—in the original "*Viderint, qui Stoicum, et Platonicum, et Dialecticum christianismum protulerunt*" (Tertullianus 1834, p. 9). In fact, he was indebted to Stoicism not only in metaphysics, but as he seeks to show Boersma (2014), also in dialectics. He did not show, at least on a demonstrative level, an affinity for philosophers. As Hager points out, Tertullianus pointed to a discrepancy between theory and one's personal life (Hager 1978, p. 339).

As Hladký says, Tertullian, under the influence of Aristotle, also distinguishes between accident and substance, and he understands the terms as they used by Aristotle (Hladký 2020, p. 30). As Kitzler points out, he perceives the term 'substance' as a specific thing (Kitzler 2014, p. 506), but as Kitzler's research shows, there are, although rarely, other meanings in the term's use in Tertullian's writings. The term 'substance' is used in a philosophical sense, reminds Stead (1963). The use of this term in the sense of matter is not very common in Tertullian. Other ways of using the term are unique. In *De anima*, the Carthaginian writer shows, especially in the field of psychology, despite the fundamentally different conceptions of the soul, some influences by Aristotle (of the 19 memories of Aristotle that have been preserved, 15 can be found in the aforementioned writing) but, in spite of that, as stated by Fredouille (2005, p. 16), it is not desirable to interpret Tertullian's psychology in a peripatetic manner.

### 2.3. Tertullian's Traducianism

Although it is not the main purpose of this paper, we also present Tertullian's unusual interpretation of the genesis of man in the sense of the works in which he is a supporter of corporealism. As mentioned above, he did not maintain this position throughout the whole of his creative life. Tertullian synthesizes Hippocratic and Aristotelian theories of generation (Qi 2018). The origin of life is identical in time to conception, and our theologian does not recognize the pre-existence of the soul. The female body (unlike Aristotle) is only the environment in which the seed of the evolving man is embedded. Tertullianus distinguishes between the seed of the soul and the seed of the body. The bodily and mental semen is united in one being, and it is never separated. "By the united impulse of both substances, the whole man is stirred and the seminal substance is discharged as a product of both; the body supplying fluidity of the soul, warmth" (Tertullianus 1950, p. 244). The soul originated from God's breath. The soul is not divisible in the physical sense, but it is possible to distinguish between the mental and non-mental parts. During the development of the body, the soul does not change its size in the same way that processed gold can be knocked out of a lump into the shape of a patch, and nothing is added to it (*Dehinc cum in laminam massa laxatur, maior efficitur initio suo per dilatationem ponderis certi, non per adiectionem, dum extenditur, non, dum augetur; etsi sic quoque augetur, dum extenditur*) (Tertullianus 1950, p. 267). Tertullian's approach is that all souls are the shoots (*tradux*) of the substance of Adam's soul. "The term "Adam" does not mean his own name, but human being in general, the human race of which he is a member" (Storoška 2018, p. 63). This is how the original sin is transmitted. The part present in both types of seeds is thus released from the original body and soul. This way of interpreting the origin of the soul is called traducianism. Tertullian apparently took it from the Stoics, Kitzler (2010) thinks. He also notes that the encratites had a similar idea of the origin of souls. Leal points out that the allegorical connection between immortality and its image, present in many early Christian writers, is absent in Tertullian's writing (Leal 2014, p. 23). However,

according to Tertullian's interpretation, Christ did not receive the contamination of sin from Adam through the seed, since he was born of a virgin and not of human seed, as Petrey (2014) points out. Traducianism can thus be seen as one of the alternatives to Origenes' pre-existential psychological concept. Kitzler is of the opinion that the question of the interpretation of the original sin played only a marginal role in Tertullian's reasoning about traducianism. Traducianism helped Tertullian, like corporealism, to argue against the Gnostics: "Tertullian eliminates the Valentinian trichotomic model of the human soul with its selective, emotional, and spiritual component" (Kitzler 2010, p. 380).

## 3. Corporealism and Tertullian Christology

The same position in the field of ontology and anthropology was shown by our author in another of his works—*De carne Christi*. According to current opinions, the work was created 10 years before *De anima*. We will go through selected aspects of the treatise that will be relevant to the topic at hand in terms of the context of the corporealist position. At the time of writing, as Kitzler states, there was no doubt about the deity of Christ (Kitzler 2015a, p. 44). The essence of the controversy in Christology was the different views on the body of Christ.

Tertullian argues about the body of Christ primarily with three other theories: those of Marcion, Apelles, and Valentinian Gnosis. This moment is considered pivotal. The ideological position of the Body of Christ was one of the most important problems in theology at the time. The manner of dealing with this problem influenced fundamental theology, trinitary theology, Christology, etc. to a significant degree. With respect to Christian thought, this problem serves as a demonstration of the center of Tertullian's ideological polemic aimed at his opponents. In the context of the mentioned treatise, this problem is central, and his ideological opponents are significant. What follows is a summary of the fundamental opinions of the mentioned opponents and the most significant ontological aspects of the treatise *De carne Christi*.

Tertullian begins directly with an open problem: the so-called modern Sadducees claim that Christ did not have a body, or if he did it was not a human body. It is a very old problem and, according to Epiphanius, the *Gospel of the Ebionites* had already denied the humanity of Christ. The resurrection of his human body would mean the same possibility for the human body. He names Marcion as the first ideological antipode in the open problem. Tollinton defines Marcion's dualism as two-fold: between the (reduced) Gospel and the Old Testament and between spirit and matter (Tollinton 1916, p. 263).

He denied the reality of the body of Christ (*negavit et carnem*) in order to negate his birth. He proclaims Marcion to be the father of docetism (*qui carnem Christi putativam introduxit*). Such an opinion would completely deny the birth of a virgin, etc. Tertullian claims that Marcion's attitude negates all reports of Christ's origin: "It is, I suppose, on these considerations, Marcion, that you have presumed to delete all those documents bearing on Christ's origins, to prevent his flesh being proved to be flesh" (Tertullianus 1956, p. 7). In Tertullian's argument, Marcion tries to show that, for God, the birth of a woman is something inappropriate. In an imaginary dialogue, Tertullian tells him that if he refused birth, he would not take on the form of a human being. He also argues with the Gospel of John (Jn 1: 32–34) (James 2000), where the Holy Spirit descended into the body of a dove. The incarnation of Christ into a true body with all its shortcomings is proof of God's love for man, says our writer. "For his sake he came down, for his sake he preached the gospel, for his sake he cast himself down in all humility even unto death, yea, the death of the cross" (Tertullianus 1956, p. 15). It is the incarnation of the whole reality of the Son, not just the body, as Steenberg points out (Steenberg 2008, p. 125). In Tertullian's eyes, denying the physical nature of Christ's body would mean a deconstruction of the soteriological dimension of Christ's work in the world. His strict antidocetism, according to Dunn, prevented Tertullian from recognizing Mary's virginity even after the birth of Jesus, because it would weaken the claim that Jesus was the owner of real humanity in his body (Dunn 2007)—an important theological point of emphasis that was later adopted

and promoted, especially by 16th century Lutheran theologians such as Martin Chemnitz (Valčo 2017).

As Otten writes, our author did not discuss the spiritual nature of Christ, which was beyond contradiction (Otten 1997, p. 251). Tertullian seeks to show, with polemical rhetoric, that denying the true physical nature of the body of Christ would also undermine his violent death on the cross, which Marcion does not deny, making his assumptions inconsistent. Here, Tertullian recites familiar words about the paradox of faith in the violent death and resurrection of Christ that are often quoted (*Crucifixus este dei filius; non pudet quia pudendum est...*). As Harrison (2017) and Li (2012) point out, several interpreters and writers often used the twisted meaning of the passage in question in the sense that Tertullian believed in the sense of religious faith because it was impossible. Here, a short note about an often-misinterpreted passage is in order. The reduction of the given passage to the winged phrase *credo guia absurdum* has entered into the heritage of rhetoric figures, and is often used in arguments (see, for example, Kosheleva and Kreinovich 2017). Tertullian certainly did not maintain the assertion that the absurdity of a statement makes him more credible. He states: "The Son of God was crucified: I am not ashamed—because it is shameful. The Son of God died: it is immediately credible—because it is silly. He was buried, and rose again: it is certain—because it is impossible" (Tertullianus 1956, p. 19). Kitzler (2015b) reminds us that here Tertullian works with a rhetoric tradition known from the times of Aristotle that asserts that evidence is more believable when it is less probable. It is not related to the alienation of reason with respect to faith, nor does he want to claim that irrationality is one of the criteria for faith. His use of the wide diapason of logical and philosophical arguments in his theological polemic is probably the greatest proof against the winged, but overturned, interpretation of the given passage.

Christ is both man and God. He could only become man because he had a mortal body. Suffering testifies to his humanity (*passiones carnem hominis probaverunt*). The body and the resurrection are not delusions.

In controversy, Tertullian comes to another Gnostic, Apelles. It is worth noting that modern research does not always prove Tertullian right in the credibility and accuracy of the description of the teachings of the Gnostic representatives. Researcher Clivaz points out that the reconstruction of ebionism by Tertullian, according to which an angel settled in Jesus, is relatively inaccurate (Clivaz 2010). Apelles claimed that Christ had a body that was not born. He said that the body of Christ is made of stars (*de sideribus*). According to Tertullian, Apelles cannot prove that the body of an angel, which was to be an analogy for the body of Christ, is astral in nature. Here, they claim that angels are spiritual substances, but they have a body of a specific kind. They did not have a body of flesh and bones. "It is agreed between us that the angels wore flesh not their own, seeing they are by nature of spiritual substance—though they have a body, albeit of its own kind" (Tertullianus 1956, p. 25) (...*utpote natura substantiae spiritalis—etsi corporis alicuisus sui tamen generis*). Here, it is clear that Tertullian, as a proponent of corporealism, claimed that it was a body of its own kind, since everything existing must have a body (Kitzler 2015a, p. 83).

Another consideration is interesting from an ontological and cosmogonic point of view. The Bible expresis verbis does not state where angels obtained a human-like body. According to Tertullian, they can acquire a non-substantial body. Tertullian writes: "Since then it is not reported from what source they took their flesh, it is left for our understanding not to doubt that it is a property of angelic power to take to itself a body from a source not material" (Tertullianus 1956, p. 25) (...*ex nulla materia corpus*). He thus differentiates between material and non-material bodies—here, the difference between corporealism and materialism is likewise shown, where non-material, intangible existence is simply an empty set. In an interesting thought, Tertullian goes on to say that if angels make of themselves what they are not by nature, they can transform themselves into something else even without substance (*ex nulla substantia facere*). If something arises from nothing, it changes again into nothing. Here, Tertullian reacts to the general ancient view of the eternity of the world, where, as Melissos said, nothing arises from nothing, that is, something created

from nothing easily changes back into nothing. He thus admits creation from nothing, but it does not seem to mean fully fledged creation in the given context, since that which is created from nothing, again, turns into nothing. According to Tertullian, angels could transform nothingness into a body, which he justifies in a very interesting way: changing nature is more than creating matter. Our theologian speaks directly: "Being able to change themselves into flesh, they are able to change nothingness itself into flesh. It is a bigger thing to change one's nature than to make matter" (Tertullianus 1956, p. 27) (…*plus est naturam demutare quam facere materiam*). This place seems to negate the usual ancient notion of the eternity of the world, which was taken for granted by the pre-Socratics Plato and Aristotle. At stake, however, is the biblical concept of the origin of the world, which is ontologically and cosmogonically innovative and seemed to be foreign in the ancient world. The given passage apparently supports the biblical concept of the genesis of the world.

Here, Kitzler refers to *De ressurectione mortuorum*, where, Tertullian states, according to philosophers, that the world was created by God out of something, he himself defends the position that it was created out of nothingness (*ex nihilo eum protulisse, si ear protulerat, quae omino non fuerat*). In this doctrine, Tertullian does not perceive a radical difference between creation (by God) from something and creation from nothing, because just as he can give matter some form (for the creation of the individual) he can also create an individual from nothing. From the point of view of the history of ontology, this is an extremely interesting passage. It seeks to reconcile the usual ancient idea that, to put it peripatetically, did not allow for the origin from the subject to the subject from nothing, with the biblical idea of *creatio ex nihilo*. It does so against the background of the defense of the possibility of resurrection of the body even after its physical decomposition. The idea of *creatio ex nihilo* was something truly artificial in the context of ancient thought heritage. This is also proved by the cosmogony of many Gnostics, who not only did not see the Supreme God in the God of the Old Testament and the Creator, but only the demiurge. In particular, Tertullian argued with Hermogen in this sense, as "Hermogen's cosmogonic basic thesis was based on a consistent denial of the doctrine of creation from nothing (*creatio ex nihilo*)" (Kitzler 2010, p. 356). The Carthaginian writer *Adversus Hermogenem* opposed Hermogen's cosmogony. Kitzler notes that such an interpretation was theologically unacceptable to Tertullian, as was Hermogen's further assertion that the soul has a material origin and evil is identical to matter, which is the reason for the decline of the soul. The fall of the soul into sin is not the result of the influence of an evil God (demiurge), our author argues. The reason is only the free will of the soul, states Tertullian. The soul is indivisible, it is a body, and what is irrational in it comes from the devil, embodied in the serpent.

If angels were to receive a material body, it would not be astral, but would more likely be of earthly matter. However, the body of Christ had to be mortal, so he did not incarnate into any astral body. "As he was to be truly man even unto death, he was under necessity of clothing himself with that flesh to which death belongs" (Tertullianus 1956, p. 27). Recall that the African thinker "distinguished the bodily resurrection of all people in the Ontic order—into Judgement (*resurrectio generalis, in iudicium*)—from the resurrection of the chosen in salvation—to the Kingdom (*resurrectio specialis, in regnum*)" (Szram 2020, p. 106). Finally, according to Tertullian, even after the resurrection, the body could not immediately enter the kingdom of God because it had not yet undergone a phase of transformation (*demutatio*), which, while maintaining its bodily substance, would acquire new qualities, namely immortality and inviolability. In any case, Tertullian was one of those who struggled with the literal wording of Paul's letter (1 Corinthians 15:50) (James 2000) and used other biblical texts to argue for a physical resurrection (Strawbridge 2017).

According to Tertullian, the motif of the appeal is the recoiling of matter, in the sense that it is something shameful. He writes: "But inasmuch as these Apelleasts make a special point of sheltering behind the dishonor of the flesh, alleging that it was constructed for seduced souls by that fiery prince of evil and therefore is unworthy of Christ, and therefore he must needs have got him a substance from the stars" (Tertullianus 1956, p. 33). However, according to Tertullian, Apelles still considers the body of Christ to be sinful because

Heaven and the whole world were created by an angel with a mixture of repentance. It is a confession of sin, and since it simply belongs to the world, it cannot be absolved of sin. This also applies to the body of Christ. Christ was also the inhabitant of Heaven in an earthly body. What has emerged from something bears traces of its origin: "...nothing that is derived from something else, though it be other than that from which it is derived, is to such an extent other as not to suggest that from which it is derived" (Tertullianus 1956, p. 37). So, the body carries traces of two of the basic elements (earth and water).

The original form of Valentinian Gnosis can be partially reconstructed from the 'Valentinian doctrinal letter' (Chiapparini 2014). Valentinian Gnosis interpreted some passages diametrically differently than, for example, Ireneus did. As Molodet (2016) points out, this is because its representatives have completely ignored the apostolic tradition. The Valentinians also did not consider the body of Christ to be matter in the sense of earthly matter. "The Valentinian doctrine that Matter, and subsequently the world, originates from the passions of Achamoth struck the polemical mind of Tertullian" (Thomassen 2000, p. 3). The Valentinians approached this question differently because they were not united. Western Valentinians considered the body of Christ to be psychic, while the Eastern ones considered it to be pneumatic. Tertullian argues against both concepts. Although he uses the names of the founders of various Gnostic and heretical directions as representatives of the same name for the whole group, as Robertson (2018) points out, he at least differentiates between the western and eastern branches of Valentinians. It is certainly interesting that he argues again in the spirit of corporealism: if Valentinians consider the soul to be invisible, they prove that it is bodily (*corporalem*) because it has something invisible. Notice the difference: it does not denote the soul as material, but bodily. "If it does possess something by which it exists, this must be its body. Everything that exists is body of some kind or another. Nothing is incorporeal except what does not exist" (...*Omne quod est, est sui generis, nihil est incorporale, nisi quod non est*) (Tertullianus 1956, p. 43). It is an exemplary manifestation of corporealist ontology that does not need in-depth analysis. This attitude has obvious soteriological consequences for Tertullian. "The traditional pagan concept, in which the dichotomy of body and soul is very pronounced, does not make it possible to establish an anthropology in which the individual as a unit lives and is thus judged, nor does it establish an ethical perspective defined in the writings" (Moreira Alaníz 2013, p. 64).

However, another interesting argument has anthropological rather than ontological aspects. The body and soul of Christ are considered by Tertullian to be two different substances. Gros succinctly expresses Tertullian's overall argument apophatically: "as evidence of his flesh being human and derived from man, not composed of spirit, any more than it is composed of soul or of the stars, or is imaginary" (...*non spiritualis, sicut nec animalis nec siderae nec imaginariae*) (Tertullianus 1956, p. 53). In short, without the body it could not be Christ: "without doubt the same flesh upon which they wrought their savagery, for without it he can neither be nor be recognized as himself" (Tertullianus 1956, p. 81). Tertullian's efforts in this context were precisely characterized by Hackett: "Tertullian tries to feel the body weight of the incarnate Word throughout its strength and makes his weight a defense against all our attempts to angelicize the incarnation" (Hackett 2015, p. 143).

## 4. Tertullian's Controversy with Illustrative Opponents

An important aspect of this study is Tertullian's polemic with his ideological opponents, mostly in the context of his treatise *On the Flesh of Christ*. These opponents are the abovementioned Marcion, Apelles, and Valentinian Gnostics. This section focuses on their notions on the Body of Christ and cosmogony because they make up the central points, which are obviously connected to many different areas of theology. Many different theological problems are connected to the nature of the Body of Christ and the cosmogonic question of the origin of the universe. Tertullian's opponents differed significantly in their response to these problems. Their opinions on the nature of the Body of Christ are closely related to soteriology and Christology. The ordering of the world by a demiurge is, once again, a typically gnostic claim that represents gnostic dualism in various aspects, most

notably in the fundamental contradiction between the Old and the New Testament. This section also deals with selected passages related to the origin of the soul, which serve as a background for illustrating to what extent the corporealistic ontological position is an appropriate Tertullian's argument against his opponents, which will be featured in the discussion part of this paper.

### 4.1. The Position of Marcion

It should be noted that there is also the work *Carmen adversus Marcionem*, which was written by an anonymous author who published it as the work of Tertullian. In fact, it is false and originated in the first half of the 4th century. As Montero (2013) notes, despite the inaccuracy of the text, its Christology is compatible with Tertullian's and is correct.

Information about Marcion has been preserved thanks to records of his ideological antipodes. Unfortunately, some of his own works have not been preserved, although he inadvertently became a catalyst for the development of theological thought (Foster 2010). According to the available information, he was the author of two works. One is *Antithesis* and the other is a canon of biblical writings. It is interesting to us that Marcion distinguishes between the Creator—the God of the Old Testament—and the God of Jesus, who is unknown. It is a great pity that Marcion's work has not been preserved at all and we can only reconstruct it from secondary sources. The canon of Marcion, as is well known, struck out the Old Testament, considering Paul to be the only true apostle. He sees the New Testament very selectively as a conglomerate of the reduced Gospel of Luke along with the letters to the Galatians, the two Corinthians, the Romans, the two Thessalonians, the Colossians, Philip, and the Philippians. He considered the original apostles to be false deceivers who did not understand the message of Jesus. This was the reason for the election of Paul of Tarsus as an apostle.

First of all, it must be stated that Marcion was not a Gnostic, and the ecclesiastical authorities called him a heretic. He drew the consequences from the disputes between Paul and Peter, and in an effort to avoid Judaism he established the first great heresy. He claimed that the apostles outside Paul had inserted Jewish ideas into Paul's letters, which he sought to eliminate. It is interesting to reconstruct the main features of Marcion's doctrine, which is possible mainly thanks to Tertullian's critique in his most extensive work, *Adversus Marcionem*. There is currently controversy over whether Tertullian read a Latin translation of Marcion's work or translated what he needed into Latin (Lang 2017; Roth 2009). In connection with the present work, Tertullian describes Moreschini as a Christian sophist who uses the philosophical tools of Middle Platonism as needed in theology (Moreschini 2017). As we have mentioned, the Old Testament is negated and the God appearing in it is not the supreme God. He is an unknown God and Christ is his son. According to Tertullian, he justifies this dichotomy as follows: "an evil tree that produces bad fruit—in particular, bad things in general—and assumed that there must be another god to correspond with, a good tree that bears good fruit" (Tertullianus 1972, p. 7). Tertullian has an objection to the new God—all the new gods are false. With a certain dose of sarcasm, he states that, like Marcion, he was inspired by Epicurus' idea of distant unknown gods. The task of the evil God was to create the material world. However, the perception of the Old Testament and its God as evil is not anti-Semitism, because the attacks are not directed at the Jews but at the God of the Old Testament (Poorthuis 2016). Marcion considers him an ignoramus who denies that there is an even higher God above him. In this creation, it must be said that it is a creature in the manner of Plato's demiurge, not a *creatio ex nihilo*. As Tertullian writes, "if he also constructed his world from some inferior material, unforgotten and uncreated and contemporary with God—which is Marcion's view of the Creator—you must add this to the majesty of this space, that it surrounds two gods, God and material: for even material is a god who is as characteristic of a deity, unforgotten, uncreated, and eternal" (Tertullianus 1972, p. 41). Tertullianus was well aware that Marcion did not reject the common idea from ancient culture of the eternity of the world in the sense of the existence

of the present as a whole (φύσις). According to McGowan, Marcion apparently despised some order of creation (McGowan 2001).

Since Christ cannot be diverse in nature, he is completely different from the material dimension. Matter is irreparable and so he could not have a body, but only a form of delusion (Prosperi 2018, p. 130). There is a discontinuity between the visible and the invisible; the unknown God is from the invisible, and the malicious God of the Old Testament is from the visible. The incarnate Christ is only a phantom of the deity. Christ was not allowed to mix with matter.

Marcion excludes the resurrection of the flesh, with Christ as well as the dead. Only the soul should be saved. Tertullian mentions their position on this question: "Markionites, whom the apostle John declared to be antichrists, who denied that Christ came in the flesh" (Tertullianus 1972, p. 191). He argues that even the angels who conversed with Abraham and Lot were not of the supposed body. According to Marcion, the human body simply cannot be saved. He also denies the birth of Christ. He is not corporeal, and there is no need to confuse him with the creator of matter. To Marcion, Christ is a spirit and, as Tertullian points out, he is not capable of suffering. Goldstein and Stroums believe that the phantom form of Christ may have an ideological origin in Marcion's Greek concept of εἴδωλον, which can be traced back to the Odyssey (Goldstein and Stroumsa 2007, p. 439). Diogenes of Sinope also uses this term, reminds Prosperi. Diogenes of Sinope, by transforming εἴδωλον into a specific type of metaphysics, took Plato's image of psychophysical dualism. Another type of existence was to be εἴδωλον. Tertullian sought to disprove such a phantasmatic nature of Christ as soteriologically meaningless. Prosperi concludes that, according to Tertullian, this kind of existing Christ has no justification in the ontology of early Christianity (Prosperi 2018, p. 140). Interesting is Tertullian's statement that man consists of two substances, body and soul, and that the whole man should be saved, not just one substance. "If the whole man was lost, in respect of both substances, then the whole man must of necessity be brought to salvation, and there is an end of that opinion of the heretics who say the flesh finds no salvation. And besides, there is confirmation of the fact that Christ belongs to the Creator, since in full accord with the Creator he promised salvation of the whole man" (Tertullianus 1972, p. 475). His other antidocetist argument is that if, after the death of Christ on the cross, the spirit left the apparent body, then nothing would be left on the cross. "It only remains for heretical presumption to say that a phantasm of a phantasm remained there" (Tertullianus 1972, p. 503). Jesus himself tells the apostles after the resurrection that there is no delusion, that the spirit does not have bones. The body of Christ clearly consists of flesh. "So then the body of Christ it can even be argued is a body, though not necessarily flesh. And yet, whatever that substance may be, seeing that he expressly says the body of him who, he goes on to say, has risen again from the dead, 'body' must of course be taken to mean a body consisting of flesh, the flesh against which the law of death has been pronounced" (Tertullianus 1972, p. 597). According to Tertullian, the bodily substance of the body of Christ should be similar to the sinful body, but without sin. Instead of redemption through suffering, the phantasmatic and illusory Christ would actually deceive death. "For he would not have piled on the horror, lifting on high the virtue of subjection, if he had known this to be imaginary and phantasmal, if Christ had cheated death instead of suffering it, and in his passion had performed an act not of power but of illusion" (Tertullianus 1972, p. 639). On the one hand, Marcion accepts Christ's suffering and death; however, on the other hand, he is a proponent of docetism. This traditional statement by Tertullian was challenged by Wilhite (2017), who thinks that there is not enough evidence to consider Marcion a docetist. In addition, however, the information we have is very vague. Moll thinks that the second, competitive tradition from which certain information comes (Pseudo-Tertullian, Epiphanius, and Philastrius) is unreliable (Moll 2008).

### 4.2. The Position of Apelles

Apelles was a disciple of Marcion who apparently reconciled his theology with Gnosis (Hanig 1999). His work has not been preserved and the data exist only from the responses of opponents. According to the preserved information, he reconciled Marcion's dualism of the two gods by declaring the demiurge a created angel. The world was created by an angel at the command of God, whom he perceives to be monotheistic. It was not created directly by God, so it is imperfect (Matoušek 1994, p. 220). A question that arises is how much he took from Philumen, who, according to Hanig (1999), claimed that Christ was not born, but had a real body. The astral body of Christ spoken of by Apelles was his own implementation. Hieronymus spoke of writing a gospel dependent on Marcion (Greschat 2000, p. 33).

### 4.3. The Position of Valentinian Gnosis and Gnostic Writings Related to It

Allusions to and quotations from Paul (Dubois 2018) are common in Valentinian Gnosis. The Valentinians had a different approach to the body of Christ. Valentine himself "uses Jesus' remarks on the return of the unclean spirit to explain his blessing of a pure heart" (Berglunde 2019, p. 52). Western Valentinians identified the demiurge with the God of the Old Testament, as preserved in Ptolemy's *Letter to Flora*. Although the Valentinian system is inconsistent based on our limited knowledge about it, it is probably possible to compile two mythological systems, A and B (Chalupa 2013, p. 52). According to version A, the material world is a product of ignorance and will perish when the aions acquire full knowledge.

Several Gnostic texts of Valentinian provenance have been preserved. Some can be attributed directly to Valentinian Gnosis, while others only bear its influences. The texts in question document the theological moments to which Tertullian responded. In general, it can be stated that in Valentinianism, matter has a secondary, derived existence, since it arises only at the end of the cosmogonic process (Thomassen 2000, p. 4). Various texts have a considerable number of documentary features.

Valentinian texts are those that have certain features of Valentinian Gnosis. *Eugnostos the Blessed* is one of them. Its reworked version is *The Sophia of Jesus Christ*, which has elements of set Gnosis. In the Christian writings with the Gnostic elements of *Authoritative teaching*, there is the concept of the material soul. It is seemingly similar to Tertullian's corporealistic notion. *The Secret Book of John* is probably composed of several originally separate texts, while the description of the pleroma is of Valentinian origin.

*The Acts of John* is a text that, according to some opinions, was created at the end of the 2nd century. It may have, therefore, provoked Tertullian to a theological reflection. This distinctly heterodox work is influenced by Valentinian Gnosis, at least in some respects. Snyder points out that his interpretation is sometimes influenced by the analysis of canonical Acts (Snyder 2015). Apparently, doctrinal views can be found in several contexts. Christ's suffering was only apparent: "I was reckoned to be that which I am not, not being what I was unto many others: but they will call me (say of me) something else which is vile and not worthy of me" (The Acts of John 2020, p. 68). The text further states that blood did not flow from Jesus, nor was he pierced or hung on the cross. The second doctrinal passage in this work, in turn, writes about his body changing form, changing state, and leaving no traces. "Sometimes when I would lay hold on him, I met with a material and solid body, and at other times, again, when I felt him, the substance was immaterial and as if it existed not at all" (The Acts of John 2020, p. 66).

*The Apocryphon of John* speaks of the changing appearance of Jesus, sometimes looking like a child, other times like an adult man. Jesus himself said in this book, "I am the Father, I am the Mother, I am the Son" (The Apocryphon of John 2000, p. 19). The monad is described in this Gnostic work as neither physical nor without a body (The Apocryphon of John 2000, p. 77). This moment is interesting, and it cannot be ruled out that Tertullian could have defined himself to be against apophatically expressed mutually oppositional qualities by a clear cataphoric statement about the corporeality of God. Jaldabaoth plays

the role of demiurge here, similar to *Timaeus*. As Vítková points out, the celestial man, Adam, originated as a prototype soul from aions. In the context of the writings, it is only a prototype soul for Adam, not a universal soul for man. He was created as a divine being by emanation. In the creation of Adam, there is also his mental body, which was created before the physical body. Adam's mental body was created by archons and their angels, the lower beings who were indebted to the demiurge. There are daimons operating in it, which in the Gnostic context must be perceived as neutral. As Vítková points out, the relationship between the prototypes of souls and earthly people is not clear (Vítková 2017, p. 30). Our author also draws attention to the inspiration of the author of the work *Timaeus*, stating that it is not a matter of creating a physical body, but of creating a mental body, originating from the angels of the demiurge. Adam later also has a physical body. Vítková believes that, in the case of the mental body, it is a light substance by which the mental body differs from the physical body, which is made from a heavy substance (Vítková 2017, p. 78). Demiurgos threw him into the physical body of the four traditional elements when he realized that he still could not find anything wrong with him. The theology of this work is typically Gnostic, offering a model that represents the transcendent God of man, whose divine substance is protologically given to the people created in his image (Litwa 2014).

Something can also be found in the periodical reference literature. Before the discoveries in the 20th century, one of the main sources of information on Gnosis and heresies, Hipollytus, in the third chapter of the eighth book, mentions the specific origin of the apparent body in docetists. At the baptism in Jordan, "he received a figure and a seal in the water of (another spiritual booty beside) the body born of the Virgin" (Refut. VIII, 3) (Hippolytus 1886, p. 120). The body obtained in this way in water was later nailed to an accursed tree.

The reference literature has also been preserved regarding Valentinian Gnosis. Clement of Alexandria (Clement of Alexandria 2016) left behind a short work called *Excerpta ex Theodoto*. This is a set of Clement's notes, which relate mainly to the named Valentinian Gnosis. This work seems to distinguish several kinds of bodies, saying that spiritual bodies will be saved. "The angels are bodies; at any rate they are seen" (Clement, Excerpta, 14) (Clement of Alexandria 2016, p. 37). The most interesting thing is that Theodotus also considers the soul to be a body of a kind in reference to Clement. If his extracts correspond to the views of this Gnostic, it could also have been a point of inspiration for Tertullian. Several scriptures speak of the spiritual body. According to this text, Jesus put on the psychic body of Christ. This body was naturally invisible and could function in the sensory world only with the help of divine preparation (Clement, Excerpta, 59) (Clement of Alexandria 2016).

In the Tertullian context of the problem we are looking at, it is worth mentioning the parallel warning of the almost identical writings *The Blessed Eugnostos* and *The Sophia of Jesus Christ* about chaos as a stage ahead of the order. *The Blessed Eugnostos* simply mentions the revelation of chaos, following the time of the Unborn, and also speaks of the worlds belonging to the heavens of chaos. Interestingly, the author uses the plural "worlds", as well as talking about the heavens of chaos. Their worlds seem to mean material chaos. The mythology of the writings is influenced by the Egyptian religion, but not to the extent that the mythology of the Ophites is (Gaston 2015). *The Sophia of Jesus Christ* mentions, in chronological order, the revelation of chaos, which preceded the creation of forms in the worlds of the heavens of chaos. It also brings as a new element the creation of forms, i.e., arranged entities with a shape to be created in chaos, i.e., to replace chaos with arrangement. From the point of view of Gnostic directions, the hard-to-classify *Protenoia in three forms* also mentions chaos as a cosmogonic stage.

Interesting moments, significant directly in connection with the possibility of a Tertullian reaction, are present in the *Authoritative teaching*. This work contains numerous Gnostic elements, while some of the paintings used have features of Valentinian provenance (Oerter and Vítková 2017, p. 102). The temporal classification of the origin of the work in the 2nd century allows for the hypothetical assumption that Tertullian could have known

it. The specific motive to which our author could have reacted concerns the psychology of the writings. The spiritual soul was thrown into the body and "it became a brother to lust (ἐπιθυμία) and hatred and envy and a material (ὑλική) soul (ψυχή)" (Authoritative Teaching 1979, p. 263). The body comes from desire and desire from the material substance. The material soul became their brother, that is, the spiritual soul degraded. In an allegorical interpretation, the author describes the decline of the soul into a state similar to that of an animal. The material soul and the spiritual soul are thus two kinds of soul, while the material soul means a connection with desire, evaluated very negatively. For the spiritual soul, it is a regression to become the brother of the material soul. We believe that it is possible to agree with Oerter and Vítková (2017) that the author of the work did not mean the Platonic division of the soul into parts that were subsequently taken over by Middle Platonism. In Augustine's references to *De regressu animae*, there is nothing to suggest that Porphyry took the "spiritual soul", which he seems to have primarily spoken of as the creator of a doctrine of salvation (van den Broek 1979, p. 262).

The difficult-to-categorize *The Second Treatise of the Great Seth* also bears certain Valentinian elements. *The Second Treatise of the Great Seth* is characterized by its eclectic use of different Gnostic traditions, such as Sethian, Valentinian, and Basilidian, setting it apart from any clear categorization (Jiménez 2020). It contains specifically docetic passages. Christ secondarily entered the body of his choice. His death at the crucifixion did not really take place: "I did not die in solid reality but (ἀλλά) in what appears" (The Second Treatise of the Great Seth 1996, p. 163). According to the treatise, they died his death, and they nailed their own man to the cross. They killed only the body created by the archons themselves. Christ says in this scripture that he did not drink bile and vinegar. It does not deny the reality of the body of Christ, but its humanity and suffering. He entered the body temporarily and was not in it at the time of the torture, so he actually deceived the archons. Similarly, the reality of Christ's suffering is denied by the Gnostic *Peter's Letter to Philip*. As van Os (2012) points out, the designation of the apostles in this document does not serve to identify the Gnostic group but is didactic. In it, the author puts a stern remark in Peter's mouth, according to which Jesus is foreign to suffering. There is a noticeable perception of the body as something bad, which is typical of Middle Platonism. Let us just recall that even the *Conversations with the Savior*, which bears certain features of spiritualizing predictive thinking, says that one cannot see a place of pure light without evil as long as one has a body.

It is interesting that the *Gospel of Philip*, the Gnostic Valentinian work, has a specific relationship to the body. In it, the author looks at the problems from several angles at once, which complicates the interpretation. "Despite its heterogeneous contents, the Gospel of Philip expresses themes that bring it into contact with the canonical gospels" (Jacobi 2018, p. 77). Metaphorically, the body of Christ is considered by the author of the gospel to be his word, and his blood is the Holy Spirit; however, in the end, he speaks of the human body. For clarity, we will present the whole paragraph: "Some are afraid lest they rise naked. Because of this they wish to rise in the flesh, and they do not know that it is those who wear the flesh who are naked. It is those who [...] to unclothe themselves who are not naked. "Flesh and blood shall not inherit the kingdom of God" (1 Co 15:50) (James 2000). What is this which will not inherit? This which is on us. But what is this, too, which will inherit? It is that which belongs to Jesus and his blood. Because of this he said "He who shall not eat my flesh and drink my blood has not life in him" (Jn 6:53) (James 2000). What is it? His flesh is the word, and his blood is the Holy Spirit. He who has received these has food and he has drink and clothing. I find fault with the others who say that it will not rise. Then both of them are at fault. You say that the flesh will not rise. But tell me what will rise, that we may honor you. You say the Spirit in the flesh, and it is also this light in the flesh. (But) this too is a matter which is in the flesh, for whatever you shall say, you say nothing outside the flesh. It is necessary to rise in this flesh, since everything exists in it. In this world, those who put on garments are better than the garments. In the Kingdom of Heaven, the garments are better than those that put them on" (Gospel according to Philip

1989, pp. 153–55). In the paragraph, he speaks of the body as despicable, although there is something rare hidden in it: the soul. He also allegorically speaks of new garments in the kingdom of heaven, which will surpass those who wear them. There is a Spirit in the body, and logos enters the body as well. The body of the Lord is real and perfect, whereas the human body is imperfect, as it is an image of the real body. It is reminiscent of the relationship of ideas and their images in things in *Timaeus*. Christ is a perfect man (Gospel according to Philip 1989, p. 157). On the other hand, Christ did not appear as he is, but to be visible, e.g., people as men and angels as angels. Although van Eijk (1971, p. 95) points out that it is difficult to ascertain what the author means by the body, we believe that the text of the Gospel is not denying the meaning of the reality of the body of Christ, but that the meaning expresses a different sense than the substance of his body. This is not a delusion, as some docetists claim. Marking the body of Christ as perfect could evoke its belonging to a pneumatic body, which would correspond to the ideas of eastern Valentinian Gnosis, but this is only a hypothesis. It is also worth mentioning the analogy of glass decanters that are made of fire. "But if glass decanters break, they are done over, for they came into being through a breath. If earthenware jugs break, however, they are destroyed, for they came into being without breath." (Gospel according to Philip 1989, p. 167). It is an analogy of the soul and the body. The soul was created by the breath of God; therefore, it is renewable, which is not true of the material body.

A few comments need to be made. *The Gospel of Philip* is esoteric, pointing to the difference between perfect and imperfect Christians (Twigg 2015). The allusion of the text to Mary of Magdalene, who became known as Jesus' companion, is also well known. In this context, it should be recalled that other fragments of other writings on this topic were proven to be forgeries (Vítková 2019; King 2013; Vítková 2013). Interesting also are the reflections of Hartenstein (2009) that, according to certain references in *The Sophia of Jesus Christ* and *Epiphanius*, there should be another gospel associated with Philip.

The Valentinian *Gospel of Truth* mentions the pre-existence of souls. The body is likened to vessels that people break after corruption. Although the text does not state this directly, the author does not seem to count on a physical resurrection. Kreps claims that this work functioned as a gospel internally for the Valentinian community (Kreps 2016). *The Expository Treatise on the Soul*, a work with Valentinian elements, also clashes with the pre-existence of the soul, which, if it enters the body, "falls into the hands of many robbers" (The Expository Treatise on the Soul 1989, p. 145). The resurrection concerns only the soul.

*The Book of Thomas the Contender* is an early Gnostic work from the period of the origin of Christian Gnosis, so it cannot be included in a specific Gnostic current. This work emphasizes the search for the soul's own identity (Hartin 1999). Due to its period of origin, Tertullian could have come into contact with it, so we make an interesting connection. Corporeality is evaluated extremely negatively. The human body originated from physical contact in the same way as the bodies of animals, so it has an extremely low degree of validity: "… [the] body is bestial" (The Book of Thomas the Contender 1989, p. 185). The body connects the soul with lust and thus diverts it from knowledge and salvation. The body is volatile and changeable. Blind people dependent on pleas are equal to animals. A similarly evaluated image of the material world, expresis verbis but not naming the body, is offered by the Valentinian *Tripartite Tractate*.

*The First Apocalypse of James*, which adheres to the position of docetism, also has Valentinian features. In this work, Christ denies his suffering post factum. "I did not suffer in any way, nor have I been distressed. And these people have done me no harm "(The First Apocalypse of James 1979, p. 83). In a similar manner, the *Apocalypse of Peter*, known for its violent fantasies about the suffering of the wicked in the grave, describes the suffering of the body as apparent, as pointed out by Gilmour (2006). The author of the work writes: "He whom you see above the cross, glad and laughing, is the living Jesus. But (δέ) he into whose hands and feet they are driving the nails is his physical part (σαρκικόν), which is the substitute" (Apocalypse of Peter 1996, p. 241). The demons crucified the son of their glory. The non-corporeal body of Christ was delivered. Christ, therefore, has other kinds of

bodies, both corporeal and non-corporeal. In addition to the non-corporeal body, Christ is, according to the author, the spirit of reason. *The Second Apocalypse of James* also refers to the body as something sinful.

## 5. Influence of Stoic Philosophy

This section will briefly examine how Tertullian was inspired by the Stoics. This fact is well-known. Here, Stoic logic most likely played a role, as it constitutes a more robust system than Aristotle's subject-predicate logic. Lopez-Astorga claims that Stoic logic is not only a particular constatival logic, but the Stoics addressed at least general conditionals (Lopez-Astorga 2016). Apparently, Tertullian was greatly influenced by Stoic metaphysics. Stoa speaks of the corporeality of the soul. Passive is the substance without quality or matter, and the active principle is deity—reason. The substance is the first matter of all existing things (Stobaeus I, 11, 5) (Gaisford 1823, p. 265). The Stoics identify God with matter, or rather God is an inseparable property of matter. That substance is not subject to extinction, and it is divisible. The corporeality of the soul in the thinking of Zeno is clearly reported by Tertullian himself (*De Anima* 5) and claimed, e.g., by Stobaeus (*Eclogae* I, 49, 33) (Stobaeus 1823). Tertullian realizes that Zeno puts substance and God on the same level (*De Praes Cup* 7) (Tertullianus 2015). According to Hippollyt's claim about Zeno, God is the purest body (*Philosophumena* 21, 1) (Hippolytus 1886).

According to Chryssip, in the reference of Alexander Aphrodisiensis, there are two principles: matter and God. God is mixed with matter, penetrating it and rendering it (*De mixtione*, 32) (Alexander of Aphrodisias 2018). Plotinus says about Chryssip's opinion, that God owes his being to matter (Plotinus *Enneades* VI, 1, 27) (Plotinus 1992). Nemesius reports on Chryssip's view of the corporeality of the soul (*De natura hominis*). Similarly, Servius speaks of our philosopher's view that God is corporeal (*Virgil Aeneid* VI, 727) (Servius 2004). That Tertullian was inspired by Stoic philosophy is well documented, as evidenced by his quoting of Stoics themselves.

*Pneuma* is a creative fire in Stoicism whose active principle is nature, i.e., God. The thing that ensures the wholeness of being is the divine mind inserted into being (Celikkol 2019, p. 1241). The substance, the substrate, never exists in stoicism without qualitative determination, because it is constantly permeated with *pneuma*. The distinction between substance and *pneuma* is equal to the difference between body and soul, but the human is substantially uniform (Long 1986).

Tertullian preferred Stoic metaphysics to other systems. He did not have too great a choice of original metaphysical systems. Plato's metaphysics, however sophisticated, was used by his obvious ideological antipodes. Epicurus's metaphysics evoked consequences that would hinder Tertullian's argumentation rather than facilitate it. Epicurus's theologia naturalis on distant material gods was too remote from the Christian notion of the Creator. Neither would his idea of the mortal soul made up of atoms be of much help to Tertullian. Aristotle's metaphysics presented similar problems. The metaphysics of matter and form was a problem for the survival of the soul independent from the body, as well as for the considerable incompatibility of Aristotle's Prime Mover with the Christian notion of God.

Tertullian had chosen Stoic metaphysics from all the existing metaphysical systems because it contained the smallest number of problematic consequences. Some of its propositions proved to be helpful in Tertullian's argumentation against his opponents. Chryssip and Zeno both identify God to be the active principle (λογός). According to some indirect interpretations (e.g., that of Eusebius), the active principle is bodily, which also carries within itself the advantage of explaining God's presence. The passive principle is material; here, the interpretations are in accordance with one another. When the divine logos permeates matter, Karfík terms it the deification of matter (Karfík 2007, p. 153). If the active and passive principles are given bodily form, or rather if the divine logos permeates matter, it implies the ontological rehabilitation of matter, which is a great argument against various kinds of Gnostic dualism. For Tertullian, it means the potential to operate with the unity of the body and the soul, as well as with the ontological unity of God and the material world.

The diapason of metaphysics that existed in the given cultural space-time was simply too limited, and Tertullian had chosen that which, from the point of view of theologia naturalis, constituted the foremost basis for the potential polemic with ideological opponents.

## 6. Discussion

This study mainly pondered the reason for and the origin and meaning of Tertullian's corporealism regarding its use in patristic thought as a relatively unconventional ontological solution. Its origin can be clearly identified in stoicism. The opinions of Tertullian's ideological opponents, mainly regarding the Body of Christ and the origin of the world, were also a part of this study. The struggle for the Body of Christ is not self-serving because the fight with docetism has soteriological consequences. Other opinions, such as those on the origin of the soul, insofar as they are connected to Tertullian's ontology, were also noticed. Traducianism is argumentatively helpful against Valentinian trichotomic psychology. The corporeality of the soul helps with arguing in favor of the salvation of the soul. A corporealist ontology also helped Tertullian explain creatio ex nihilo, which is, of course, compatible with biblical texts, unlike the gnostic notion of order created by the demiurge.

Stoic philosophy's influence on Tertullian is an obvious and proven fact. Tertullian himself refers to the Stoics in several of his works. The belonging of God, the soul, the whole earth, and the cosmos to one substance is what can be stated both about the representatives of Stoicism (especially Zeno and Chryssipus) and about Tertullian. The source of motives and attitudes in Tertullian corporealism cannot be reduced to Stoicism alone.

Several other indicators that shaped Tertullian's attitudes are also important. Many forms of emerging Christianity have also been transformed into a number of Gnostic groups. With few exceptions, their statements were fundamentally different from the nascent current of the general church. Most of Gnosticism bore the clear influence of Platonic philosophy, whether influenced directly by Plato or by various currents of Middle Platonism. Platonic dualism was certainly not an accepsection metaphysical substrate for Tertullian, as it paved the way for some Gnostic speculation. Our author himself understands some terms in an Aristotelian way, e.g., accident and substance.

Platonic philosophy has some aspects that were very attractive to the representatives of Gnosis. The dualism of matter and ideas evoked other possible dualisms, such as the dualism of the false and unknown god and the dualism between God and the world (Culdaut 1999). The consequence of dualism is also the docetism of the Gnostics, and from a modern point of view, heretical Christian currents (in our context, especially Marcion). The body is perceived to be obviously ontologically underestimated. The soul and only the soul can be the object of salvation. Even with Christ, his body cannot be deified, as something material and changeable cannot be an integral and inseparable part of Christ, no matter how Christologically Gnosis takes it.

For Tertullian, the real body of Christ is, first and foremost, an argument against the claim that only a soul without a body will be saved, as well as against the denial of the resurrection. In an effort to deny the serious ontological difference between body and soul, he also declares the soul to be corporeal. This thus avoids further dualism and denies the Gnostics' claim of a special divine spark in the soul and the Platonic triadic structure of the soul, which was altered by the Valentinian Gnostics. Above all, it rehabilitates the body and matter. The sin was not identified with matter or salvation with knowledge by the nascent mainstream of the church, like most Gnostics. The real body of Christ is not something apparent or changeable, as we have had the opportunity to see in various texts and references. Apparently, Tertullian was confronted with the work of Marcion that has not been preserved and probably had the opportunity to read the part of the Gnostic writing that advocated for docetism. He had also known Apelles' concept of the astral body. He was acquainted with *The First Apocalypse of James* (1979), a Gnostic text with Valentinian elements, which stands in a docetic position. As we have seen, the *Apocalypse of Peter*, the *Second Treatise of the Great Seth*, *Peter's Letter to Philip*, and the *Acts of John* also show signs of docetism. *The Secret Book of John* reports on the changing body of Jesus. The Valentinians

considered the body of Christ to be psychic or pneumatic. Tertullian's consistent support of the material body of Christ is a position that opposes various forms of docetism, against the Gnostic radical denial of the ontological and axiological status of the body. It supports the claim of the salvation of the whole person, including the body.

Several kinds of dualism (the dualism of the body and the soul, etc.) have been contemplated by most Gnostics. The bodily God is not diametrically ontologically different from nature or from the body, which makes it possible to deny the idea of the creation of the world by a false god (demiurge, angel, etc.) of ontologically worthless matter. It is also an argument against Apelles (if God is corporeal himself, why think that Christ has an astral body) as well as against Marcion's dualism. At the same time, it is a counterattack against the identification of evil and matter, which is a significant element for most Gnostic directions. It can be an argument against, e.g., Hermogen, who advocated, among other things, for the identification of evil and matter. The corporeality of God, although not in the sense of a material body, could also be an argument against docetism, since God is not a delusion; on the contrary, it can be characterized as a body (*corpus*).

The relationship to the body is generally negative for some of Tertullian's opponents. In addition to Marcion is Apelles, who says that the body cannot be saved because of its sinfulness. *Authoritative Teaching* also perceives the body to arise from desire. *Conversations with the Savior* completely excludes the body from seeing pure light without evil. The body is also perceived with negative connotations in the Valentinian Gospel of Truth and *The Book of Thomas the Contender*.

On the one hand, the corporeality of the soul is an argument against those who want to diametrically ontologically differentiate the body and the soul with the anticipation that the body cannot be saved. On the other hand, it helps to explain inherited sin in a specific way through traducianism, thus creating an argument for the non-subordination of Christ's soul to the law of inherited sin, since he was born of a virgin. He may have had reports on the material soul from the *Authoritative Teaching*, and the corporeality of his soul was well known to him from Stoicism. It is possible that he read the *Authoritative Teaching*, but even then, its influence him could only be marginal. He adopts the concept of the corporeality of the soul and the ontological unity of the soul and the body from Stoicism.

Tertullian also copes with the concept of the origin of the world. The concept of *creatio ex nihilo* was essentially foreign to the ancient world of thought. Ancient philosophy was based on a cosmogonic picture of the universe, which had its origin in chaos, or at least in something. This cosmogonic idea was also taken over by the majority of Gnostics, who to a greater or lesser extent flirted with Christianity. This can be said of the vast majority of Gnostic texts with which Tertullian may have come into contact. Even his ideological antipode Marcion did not include *creatio ex nihilo* in his interpretation of the origin of the world. Apelles also counted on the demiurge, explicitly excluding God's contribution to the creation of the world. Jaldabaoth acts as a demiurge in the *Secret Book of John*. *Eugnostos the Blessed* and its sister text *The Sophia of Jesus Christ* speak of the chaos that precedes order. The situation is similar in *Protenoia in three forms*. The solution to the problem of the creation of the world out of nothing was original. Tertullian treats the possibility of the creation of the world out of nothing without talking about a possible alternative. Related to this is his reflection on the creation of a body by angels even without substance. In this context, corporealism allows for the emergence of a material or non-material bodily entity, which in the context of a possible explanation of cosmogony in the spirit of *creatio ex nihilo* appears to be more flexible than Platonic idealism or materialism (e.g., that of Epicurus).

As we have mentioned, Tertullian was not a consistent philosopher, much less a metaphysician. The ideological dividing line did not lead primarily through philosophical fields but through theological fields. The lines of Tertullian's thought were dominated by theology, and philosophy was only an aid.

The effects of Stoicism, or of some ideas of epicureanism (the corporeality of the gods), on Tertullian meant grasping a concrete form of the means of expression and religious postulates, especially in confrontations with ideological opponents. In his thinking, philos-

ophy had a concrete form of the realization of selected ecclesiastical doctrines and provided argumentative support to his opponents. Arguments, as is well known, were often used purposefully by Tertullian. His philosophical attitudes must, therefore, be understood as subordinate to his theological positions. The zealous speaker and skillful argumentation strategist used argumentation tools tactically: "Disentangling Tertullian's convictions from his rhetoric is notoriously difficult" (Wilhite 2021, p. 490). The determining cultural influence on Tertullian was not philosophy, but Christian theology. Stoic philosophy, as an important cultural phenomenon that has penetrated every geographically Hellenized area, is thus present in Tertullian's work in close accordance with his theological attitudes.

## 7. Conclusions

In this study, we tried to identify possible contextual historical influences that may have influenced the relatively atypical ontological basis for an early Christian thinker who swam with the ideological current of the mainstream church. It was mainly focused on *On the Flesh of Christ* and *The Treatise on the Soul.* Tertullian's ontological inspiration was identified. His opinions were confronted by the opinions of his ideological opponents in connection to a direct polemic. The focus was put mainly on the aspects of the ontological nature of the Body of Christ and the origin of the world. The topic of traducianism in connection to Tertullian's ontology was also touched upon. It was stated that this early Christian philosopher's choice of an atypical ontological position was influenced by the need to argue against the most significant opponents in the field of theology. Tertullian used an ontologically argumentative arsenal in order to promote his theological views in the theological polemic with his ideological opponents.

Stoic philosophy was a primary inspiration. Despite some statements in which Tertullian pretends to be independent of Stoicism as a thinker, his dependence on Stoicism in metaphysics is inescapable. He also uses Stoic dialectics.

Marcion of Sinope, later referred to as a Gnostic and heretic, could have influenced Tertullian with his attitudes. He, as we have shown, deftly chose different strategies and tactics to defend his beliefs. One of his basic beliefs is the salvation of the whole man, i.e., body and soul. The salvation of the whole man as a leitmotif shifts Platonic anthropology to a negative level. Marcion, who was more or less dependent on Plato as well as most Gnostics, showed that the possibility of manifesting the ontological unity of soul and body was advantageous for arguing in favor of the salvation of both the soul and body. The soul is perceived as a non-substantial body of its own kind. This argument was also connected with the reality of the body of Christ. The ontological unity of soul and body in man and in the body of Christ was a good path to holistic soteriology. At the same time, such an ontological basis is a good reason to argue in favor of the resurrection of the body. In addition, a relatively large number of Gnostic writings with which Tertullian could have come into contact bore the hallmarks of docetism. In any case, the strong docetic tendencies of Marcion, Apelles, the Valentinians, and some other Gnostic currents could only confirm Tertullian's use of corporealism as an argumentatively advantageous ontological position.

Another important conviction of Tertullian was the creation of the world by God, not by subordinated beings. Moreover, it was creation from nothing, and the material world that was created had an ontologically harmless character. The concept of the ontological unity of the Supreme God and the world of bodily matter helped to overcome the dualism of the Old and New Testaments (in the interpretations of most Gnostics) as well as the axiologically negative view of the material world as evil and sinful on the basis of its ontological nature and origin (evil supernatural beings).

A new opinion is also given on *creatio ex nihilo*, a concept essentially foreign to the ancient world. Tertullian's pointing to the possibility of the body being created by angels even without substance opens the door to an explanation of *creatio ex nihilo* in a cosmogonic sense. What is possible for an angel in the sense of the creation of the body is all the more possible for God in the sense of the creation of the world. At the same time, corporealism does not differentiate between the creation of a physical or mental entity, which eliminates

the problem of the creation of entities of an ontologically different nature. It also seems to be advantageous when arguing in favor of the belief in the emergence of a world from a null existence, which was controversial compared with the long-held notion of order out of chaos. In the original Hellenistic cultural thought that formed the educational face of the Roman Empire, *creatio ex nihilo* was something utterly alien. The theory developed by Greek culture only acknowledged the change in the order, and not the formation of the universe from nothing. Tertullian's texts also make clear that this model was not taken for granted. Its contemplation was the result of Tertullian's unyielding affirmative approach towards the nascent canon of the Word as an authority, as well as tradition. Tertullian thus supported this culturally anti-mainstream metanarrative. What is important is that he had shown himself to be fundamentally compatible with the nascent message of the Word and tradition, as well as argumentatively appropriate against the culturally experienced model of order created by the demiurge in the Hellenistic space, so happily utilized by most Gnostics and Marcionites.

Corporealism thus appears to be an ontological position of opinion that was argumentatively advantageous to Tertullian. Tertullian was not a principled metaphysician who would consistently be a supporter of one metaphysical system. His corporealist views were evidently inspired by Stoicism, but served, in particular, his theological argumentation to support his basic theological beliefs. They were highly contradictory to the ontological basis of most of Tertullian's ideological antipodes. In our opinion, this is the use of this ontological position of opinion, which was certainly not typical of early Christian thinking. Thus, we do not perceive him to be a representative of the Christian stoa, but more as a skillful Christian theologian who was able to use corporealism to argue for his basic theological beliefs. In the network of Tertullian's beliefs, corporealism is one of the ontological and, thus, contingent beliefs, serving to support the more basic, theologically necessary beliefs.

The effect of culture, philosophy, and religious thought on Tertullian can be summarized as Tertullian absorbing the tools of philosophy, especially those of Stoic provenance, to the extent that he needed them to argue against the nascent ecclesiastical doctrine, especially against Gnostics and Marcionites. For this reason, we in practical terms do not find in it a positive use of the philosophical tools of Middle Platonism, which belonged to the philosophical arsenal of Gnostic thought. Tertullian sought to orient himself in the cultural world of religious syncretism and, particularly on the level of fundamental theology and Christology, he came into sharp opposition with some religious currents within what, at the time, was rather heterodox Christian thought. In this context, he became a supporter of corporealism at a certain moment in time. He skillfully used this ontological position when arguing against his opponents on theological issues. Although he was not a philosopher in the true sense of the word, he was not an opponent of philosophy, as is sometimes mistakenly said; he often used its tools himself.

**Funding:** This research was funded by the College of International Business ISM Slovakia in Prešov, grant number IG-KSV-01/2020-12-33/IP (Interdisciplinary solutions for the study of selected aspects of social relations in the context of today's phenomena).

**Acknowledgments:** Thanks to Zuzana Vítková for her consultation.

**Conflicts of Interest:** The author declares no conflict of interest.

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
