# Peer review of "Corporealism as an Ontological Position and Its Involvement in the Thought of Tertullian"

_religions, doi:10.3390/rel12070534_

Round 1

Reviewer 1 Report

The article takes into account the recent scholarship on Tertullian and shows good knowledge of the relevant sources, but there is no straight line of argument that leads to conclusion. What is more, I cannot find citation of the original texts of Tertullian and reference to them. Further, many terms in English are mistaken; the author speaks of "traditionalism", while obviously meaning "traducianism", "essence" perhaps instead of "substance", "Eirenai" instead perhaps of "Irenaeus". I also noticed several repetitions e.g. regarding the difference between materialism and corporealism. The writing of the paper needs improvement in many respects. I recommend revise and resubmit.

Author Response

The author highly values the good professional review of the reviewer. The author will try to correct the terminology as shown by the reviewer, as well as to insert new direct quotations from Tertullian and to better emphasize the line of argument.

Reviewer 2 Report

The notes for author are in the attachment uploaded below.

Author Response

The author thanks the reviewer for a few minor corrections, he will implement them into the text.

Reviewer 3 Report

The text is written very well, the author has shown excellent orientation in the topic, an overview of the relevant literature as well as a good foundation in philosophy. The text has the potential to appeal to a wide range of readers.
Perhaps it would be benefitial if the author at least partially included in his explanation Tertullian's "Credo quia absurdum", which is commonly understood as a certain extreme approach to the question of the relationship between faith and natural reason. However, Tertullian in no way excludes reason from the realm of faith, nor does it even mention irrationality as a criterion of faith. On the contrary, according to him, natural reason has a place in the life of the believer, because it protects him from the mistakes of heresies and mediates to him the basic truths of the faith. Thus, "Credo quia absurdum" does not refer to the relationship between faith and "conceptual thinking," but to faith and "what the world treasures". Tertullian sees philosophers, castigating them on the one hand for being the root of all heresies, but praising them on the other for providing the Christian theologians with arguments and logical tools (e.g. modus tollens). However, I am aware of the scope limitation and do not consider this addition necessary.

Author Response

The author thanks the reviewer for the comment, he will try to respond to it in a corrected version of the text.

Round 2

Reviewer 1 Report

This is a much better version but still some issues remain. I do not think it is right to claim that "in his case, the primary emphasis was on maintaining the line of official Christianity, especially in opposition to phenomena such as Gnosis and Marcion's speech" (p. 18). I very much doubt whether an official Christianity existed at the time. Also the claim that Tertullian wanted to avoid the dualism of the Old and New Testament  (p. 17) is odd in many regards. 

The claim in p. 16 "The substance of the Stoics is that parts of it may change" is odd and unclear.

Something has gone wrong in line 955.

Tertullian's influence by Stoicism is a well known truth. The question is why he finds Stoic ontology attractive.

Author Response

Thank you very much to the reviewer for the constructive review. I will try to incorporate the comments.